# Noninvasive Monitoring Strategies for Bronchopulmonary Dysplasia or Post-Prematurity Respiratory Disease: Current Challenges and Future Prospects

**DOI:** 10.3390/children10111753

**Published:** 2023-10-29

**Authors:** Tommaso Zini, Francesca Miselli, Alberto Berardi

**Affiliations:** 1Department of Medical and Surgical Sciences of Mothers, Children and Adults, Post-Graduate School of Pediatrics, University of Modena and Reggio Emilia, 41121 Modena, Italy; 163320@studenti.unimore.it; 2Neonatal Intensive Care Unit, Department of Medical and Surgical Sciences of Mothers, Children and Adults, University of Modena and Reggio Emilia, 41121 Modena, Italy; alberto.berardi@unimore.it

**Keywords:** bronchopulmonary dysplasia, post-prematurity respiratory disease, preterm infants, noninvasive monitoring, lung oxygenation, near-infrared spectroscopy

## Abstract

Definitions of bronchopulmonary dysplasia (BPD) or post-prematurity respiratory disease (PPRD) aim to stratify the risk of mortality and morbidity, with an emphasis on long-term respiratory outcomes. There is no univocal classification of BPD due to its complex multifactorial nature and the substantial heterogeneity of clinical presentation. Currently, there is no definitive treatment available for extremely premature very-low-birth-weight infants with BPD, and challenges in finding targeted preventive therapies persist. However, innovative stem cell-based postnatal therapies targeting BPD-free survival are emerging, which are likely to be offered in the first few days of life to high-risk premature infants. Hence, we need easy-to-use noninvasive tools for a standardized, precise, and reliable BPD assessment at a very early stage, to support clinical decision-making and to predict the response to treatment. In this non-systematic review, we present an overview of strategies for monitoring preterm infants with early and evolving BPD-PPRD, and we make some remarks on future prospects, with a focus on near-infrared spectroscopy (NIRS).

## 1. Introduction

Bronchopulmonary dysplasia (BPD) lacks an objective, comprehensive, and unambiguous definition, due to its complex multifactorial nature and the substantial heterogeneity of clinical presentation [1]. The most widely used operational definitions of post-prematurity respiratory disease (PPRD), or BPD, aim to stratify the risk of mortality and morbidity, with an emphasis on respiratory outcomes [1]. Therefore, the classification of this chronic lung disease requires a compromise between early identification and the accurate prediction of long-term outcomes [1].

Both the definitions and phenotypes of BPD have evolved in recent years [2,3,4,5,6,7,8,9,10]. Advances in obstetric and neonatal medicine, such as the routine application of antenatal steroid courses, exogenous surfactant replacement therapy, and gentler mechanical ventilation strategies, have led to the improved survival of extremely premature infants, and there is an increasing trend toward offering initial life support to infants born at the lower limits of viability [11]. Therefore, BPD pathophysiology has dramatically changed in the “post-surfactant era”: there are fewer severe lung injuries caused by volutrauma, barotrauma, atelectrauma, and oxygen toxicity, but very immature lungs are affected during the early phases of development [12]. “New BPD” is the result of underdevelopment or a dysregulated lung developmental trajectory, with incidents occurring in the saccular stage or earlier, resulting in reduced secondary septation, or alveolar simplification, and a dysmorphic pulmonary microvascular network [12,13,14]. Disrupted alveologenesis is associated with an increased risk of pulmonary arterial hypertension, or BPD-associated pulmonary hypertension, which worsens the prognosis dramatically. Not surprisingly, BPD-PPRD is classified on the basis of clinical evidence of functional impairment at a time when the physiological development of the human lung should be characterized by the onset of the last stage, or alveolar stage [14]. In fact, the respiratory support modality still required at 36 weeks postmenstrual age (PMA) is predictive of long-term outcomes [1,5,6,7,8,9]. It is debated whether the prognosis can be better predicted by postponing the time of assessment to 40 weeks postmenstrual age [10]. In contrast, there is a need for objective, precise, and reliable noninvasive point-of-care tests to predict the diagnosis and classify the severity of evolving BPD at a very early stage [15].

It must be considered that, during infancy and adulthood, even the immature lungs of very-to-extremely preterm (less than 32 weeks gestational age) and very-low-birth-weight (VLBW, <1500 g birth weight) infants who do not fit the classification criteria of BPD-PPRD may exhibit characteristic abnormalities in lung structure and function [16]. Prematurity itself carries a higher risk of chronic lung disease in infancy, with a phenotype similar to that of chronic obstructive pulmonary disease (COPD) in adulthood, hence close attention should be paid to the regular monitoring and early management of respiratory symptoms and comorbidities in these children and adolescents beyond the neonatal period [17,18].

The neonatal care of infants with BPD does not include a definitive available cure and challenges persist in finding targeted preventive therapies. Currently, the most effective strategy is to optimize obstetrical care and to prevent preterm delivery [1]. However, emerging stem cell-based postnatal treatments, such as the intratracheal administration of extracellular vesicles (EVs or exosomes) derived from human cord mesenchymal stromal cells (MSCs) to prevent BPD, are being investigated (EVENEW clinical trial). Pending results from large-scale trials, a future is beginning to emerge in which innovative treatments aimed at BPD-free survival will be offered in the first few days of life to extremely premature infants or to invasively ventilated higher-risk infants. Hence there is a need for multimodal risk prediction tools to noninvasively detect BPD early, to improve and standardize assessment methods, and to support clinical decision-making. Furthermore, early predictors of the response to treatment with noninvasive methods are required.

In this non-systematic review, we present an overview of strategies for the noninvasive monitoring of infants with post-prematurity respiratory disease or bronchopulmonary dysplasia and make some remarks on future prospects.

## 2. Sources and Selection Criteria

We searched the PubMed and Cochrane databases for articles published in peer-reviewed journals. Search terms included “bronchopulmonary dysplasia”, “post-prematurity respiratory disease”, “noninvasive monitoring”, and, separately, “lung oxygenation”, “near-infrared spectroscopy”, and “preterm infants”. We focused on results published after 2010, but included some results from classic literature, consensus statements, and clinical guidelines. We excluded articles not published in English (Figure 1).

## 3. Risk Prediction Models

### 3.1. Risk Prediction Early after Birth

Clinical and demographic data can support postnatal clinical decision-making and personalized monitoring strategies early after birth. Recently, a risk prediction model and a risk scoring tool were derived from a systematic review and meta-analysis of risk factors for bronchopulmonary dysplasia [19]. These were divided into three categories (antenatal, intrapartum, and postpartum factors) and the authors selected twenty-four (24) different risk factors for analysis [19]. The meta-analysis by Yu and colleagues showed a pooled effects of nineteen (19) risk factors: chorioamnionitis, gestational age, birth weight, sex, small for gestational age (SGA), 5 min Apgar score, delivery room intubation, neonatal asphyxia, invasive mechanical ventilation (MV), days on MV, MV duration > 7 days, courses of postnatal steroids, surfactant, patent ductus arteriosus, respiratory distress syndrome (RDS), sepsis, intraventricular hemorrhage, necrotizing enterocolitis, pulmonary air leak [19]. Five (5) risk factors (maternal hypertensive disorders, antenatal steroids, premature rupture of membrane, caesarean section, retinopathy of prematurity) were not included in the final analyses [19]. The prediction model was based on 9 risk factors: chorioamnionitis (OR = 3.56, 95% CI [2.49, 5.11]); gestational age per 1 week increase (OR = 0.64, 95% CI [0.62, 0.67]); birth weight per 100 g increase (OR = 0.78, 95% CI [0.76, 0.80]); sex—male (OR = 1.46, 95% CI [1.39, 1.54]); SGA (OR = 4.78, 95% CI [3.88, 5.88]); 5 min Apgar score per 1 point increase (OR = 0.71, 95% CI [0.64, 0.78]); delivery room intubation (OR = 2.77, 95% CI [2.27, 3.39]); need for surfactant replacement therapy (OR = 3.59, 95% CI [2.90, 4.45]); respiratory distress syndrome (OR = 5.08, 95% CI [4.06, 6.35]) [19]. A BPD risk prediction scoring tool for preterm infants was derived and four risk groups were stratified according to risk score in a Chinese validation cohort (total 767 infants, of which 185 were BPD infants) [19]. The risk groups were defined as follows according to prevalence rates: Low (0.5%), Low-intermediate (5.5%), High-intermediate (67.0%), and High Risk (94.7%) [19].

This study validated an easy-to-use and low-cost BPD risk-scoring tool [19]. Despite several limitations, similar tools could play an important role in the early identification and stratification of high-risk preterm infants.

### 3.2. Antenatal Risk Prediction

With chorioamnionitis, the placental inflammatory (acute and chronic inflammation) and vascular pathways (maternal and foetal vascular disease) are predictive of BPD [20]. There is a growing interest in developing models for predicting histologic chorioamnionitis and adverse outcomes in preterm infants [21]. As a matter of fact, BPD already develops in utero, and antenatal interventions in women with a higher risk of chorioamnionitis should be investigated. Placental inflammatory and vascular pathways may also be associated with the response to postnatal therapies [20].

## 4. Biofluid Biomarkers

Currently, no biomarker-based diagnostic strategy is available, but various biomarkers and “omic” signatures (genomics and epigenetics, metabolomics, proteomics, microbiomics…) specific to BPD are being studied [22]. In research and clinical settings, there is increasing interest in comprehensive endotypic assessments, obtained from easily taken biological specimens, and noninvasive monitoring techniques that, without requiring specific expertise, predict the early development of BPD in premature infants [15].

Biofluid biomarkers have been studied mainly using umbilical cord blood, urine, and tracheal aspirate or bronchoalveolar lavage fluid samples. A review focusing on biohumoral markers was recently published in a journal of the European Respiratory Society by Cui and colleagues [15].

Inflammatory markers have also been studied in exhaled breath condensates: higher end-tidal carbon monoxide (ETCO) on day of life (DOL) 14, higher fractional exhaled nitric oxide (FeNO), and higher exhaled nitric oxide (eNO) have a significant association with BPD [15].

“Omic” biomarkers of BPD include: genomics and epigenetics (higher vascular endothelial growth factor VEGF-634G>CG alleles, lower miRNA-876-3p in tracheal aspirate), proteomics (higher chitinase-3-like protein-1, matrix metalloproteinase-9 MMP9 in urine), microbiomics (lower *Firmicutes* and *Staphylococcus*, and higher *Proteobacteria*, *Ureaplasma*, and *Stenotrophomonas* in tracheal aspirate), and metabolomics (numerous biomarkers in various specimens; note that the metabolome is also affected by the metabolic activity of the lung-gut microbiome) [15].

In summary, several biofluid and “omic” markers have been studied at different days of life; however, larger, high-quality trials integrating multimodal assessment are required for clinical application. Attention should be given to markers whose results are available early in life (≤72 h of life), when diagnosis and the early initiation of experimental therapies are more likely to increase the potential benefit of treatments. In addition, early biomarkers indicating the response to treatments should be preferred.

## 5. Chest Imaging

### 5.1. Chest Radiograph

Since the Northway definition (1967), radiographic studies have been essential for identifying BPD [3]. Among chest imaging modalities, plain radiography is the primary line and the most studied in early and developing BPD due to its wide availability in neonatal intensive care units (NICUs) and the low dose of ionizing radiation used [1].

Several abnormalities of chest X-ray (CXR) have been clearly associated with prolonged oxygen requirement, including the radiographic pattern of “cystic BPD” (with “bubbly” alterations) and “leaky lung syndrome” (with “hazy-to-opaque lungs”) [23]. The interstitial pneumonia pattern on DOL 7 and several scoring systems applied to CXR in the first week of life correlate with the diagnosis of BPD [24,25]. Even earlier, on DOL 3, an increased chest radiographic thoracic area (CRTA) is a sign of hyperinflation and has been associated with air trapping, which likely reflects ventilation inhomogeneity and, in combination with lung function (lower functional residual capacity), might better predict the presence of BPD than certain clinical data [26].

### 5.2. Lung Ultrasound

Newer than CXR, lung ultrasound (LUS) is widely used in neonatology for diagnostics, prognostics, monitoring, and prevention. It is available and accessible in the NICU, it can be performed without delay and serially (point of care ultrasound or POCUS), and it does not involve the exposure of the newborn to ionizing radiation [27].

VLBW infants with BPD have greater consolidations and evidence of pleural line abnormalities than those that are not diagnosed with BPD [27]. In VLBW infants, a semi-quantitative LUS score has been demonstrated to be a useful tool in predicting moderate-to-severe BPD, i.e., the degrees of disease most relevant to respiratory morbidity [28,29]. VLBW infants without BPD have LUS scores that increase during the first week of life and decrease thereafter, whereas, among VLBW infants with BPD, LUS scores remain elevated until 36 weeks postmenstrual age (substantially higher LUS score at admission, on DOL 7, on DOL 28, and at 36 weeks PMA). Notably, LUS scores can predict a BPD diagnosis at 7–14 days of life and can predict BPD severity at 4 weeks of life [30].

Additionally, LUS scores could be early predictors of treatment response. Failure to improve LUS scores with diuretic therapy (2 days after diuretic administration) was associated with worse respiratory outcomes [31].

### 5.3. Chest Computed Tomography Scan

Other chest imaging modalities, particularly computed tomography (CT), have been studied in infants and children with BPD, but there are several limitations to their use for early and evolving BPD, largely due to the ionizing radiation exposure in growing and developing individuals who are more radiosensitive [1]. However, as low-dose CT protocols progress, neonatal lung imaging with CT will become increasingly common.

Several qualitative, semi-quantitative, and quantitative CT scoring systems have been validated, and this imaging technique applied to established BPD can predict the severity of long-term respiratory outcomes [32]. Typical CT abnormalities include bronchial wall thickening and bronchiectasis, hyperaeration or emphysematous areas and bullae, multifocal interstitial and subpleural opacities, fibrosis, and mosaic lung attenuation. Automatic and semiautomatic methods are being studied to quantify alveolar structure with chest CT scans [33].

### 5.4. Chest Magnetic Resonance Imaging

Magnetic resonance imaging (MRI) protocols are emerging as new techniques in respiratory imaging, the impact of which is likely to be most significant in the long-term outpatient management of infants and children with severe BPD-PPRD. Ultrashort echo-time (UTE) MRI is the most studied technique [17,18].

## 6. Lung Function

The main pathophysiological basis for impaired lung function are structural abnormalities such as alveolar simplicity, a decreased number of alveoli, and enlarged alveoli [34]. Pulmonary function tests (tidal flow-volume loops, the raised volume rapid thoracic compression technique, functional residual capacity via multiple-breath washout or plethysmography) have been proposed to further investigate the diagnosis of BPD and validate the proposed definitions [1]. Nevertheless, they are not yet integrated into routine neonatal medicine due to the need for specialized skills and equipment, and there is no consensus on which technique is preferred to collect the early lung function data of the infant in the NICU in order to identify the dysregulated lung developmental trajectory that characterizes BPD [1,35].

Preterm birth and VLBW status have been consistently associated with reduced total lung capacity, lower lung compliance, increased airway obstruction and respiratory system resistance, impaired gas exchange, and premature decline in lung function [34]. Pulmonary function abnormalities in preterm newborns that predict BPD development early include: lower functional residual capacity (FRC), lower compliance of the respiratory system (Crs), and higher resistance (Rrs) on DOL 3 and 14. Other lung function parameters for which a deviation from normalcy has been demonstrated on DOL 7, 14, and 28 are as follows: higher effective airway resistance per kilogram (Reff·kg^−1^), lower tidal volume per kilogram (TV·kg^−1^), lower functional residual capacity (FRC), a lower ratio of the time to peak tidal expiratory flow to the total expiratory time (% T-PF), and a lower ratio of the volume of the peak tidal expiratory flow to the total expiratory volume (% V-PF) [15,36].

A more accurate and standardized interpretation of the flow volume loop observed on the monitor in mechanical ventilated infants could return parameters similar to those coded for the tidal flow-volume loops technique. The interruption technique, forced oscillation technique, and pulse oscillometry for the study of resistances and reactances could be implemented. Other serial measurements of lung function (spirometry, multiple breath washout, plethysmography, and other advanced pulmonary function tests), lung diffusion testing (diffusion lung capacity for carbon monoxide or DLCO), polysomnography (sleep study), and the assessment of respiratory morbidity should be part of the post-discharge follow-up of preschool and school-age children and adolescents with BPD [17,18,37].

## 7. Physiological Tests

Attempts at a “physiological definition” of bronchopulmonary dysplasia were initiated in 2003 by Walsh, who proposed a technique to standardize the assessment of BPD using a timed room-air challenge [38]. BPD was defined at 36 weeks PMA for very preterm infants who required positive pressure ventilation (PPV) or a fraction of inspired oxygen (FiO_2_) ≥ 0.3 to maintain pulse oximeter oxygen saturation (SpO_2_) at a range of 90–95% [7,38]. Most of the remaining infants, who mostly required oxygen supplementation with a low-flow nasal cannula (NC) 1–2 L/min, were considered eligible for an oxygen reduction test (ORT) [7,38]. Those who maintained SpO_2_ ≥ 90% for 30 min while breathing FiO_2_ 0.21 were classified as “no BPD” in Walsh’s 2004 “physiological definition” [7,38].

Recently, a modified physiological test for BPD has been proposed, which includes transcutaneous PCO_2_ (tcPCO_2_) monitoring [39]. After 28 days of life and at 36 weeks postmenstrual age, infants on continuous positive airway pressure (CPAP) and/or on oxygen supplementation with nasal cannula and FiO_2_  ≤ 0.30 at rest, with SpO_2_ between 90 and 96%, were considered eligible to undergo the modified physiological test, and a timed stepwise reduction of FiO_2_ and/or CPAP to room air was performed. During the test, the infants were monitored continuously, and test failure was defined as the occurrence of any of the following events: desaturation characterized by SpO_2_ between 80 and 89% for 5 min with tcPO_2_ < 50 mmHg, or SpO_2_ < 80% for 1 min. Other causes of test failure were considered apnea (cessation of breathing for >20 s) and/or bradycardia (heart rate < 80 beats per minute for >10 s) [39].

There are no standard criteria for weaning preterm infants from CPAP and/or supplemental oxygen, so physiological testing represents not only an effort to standardize the diagnosis of BPD but also a clinical tool for weaning respiratory support [1,7,38,39]. However, these are delayed tests, dedicated to infants who require low levels of respiratory support. Similar approaches to investigate the response of the respiratory system under stressful conditions in a standardized way could also be useful in the early days of life. Routine pulse oximetry, polysomnography (sleep study), hypoxia testing, and blood gas analysis are other techniques for monitoring and studying lung pathophysiology [40]. New tools for continuous pulmonary monitoring and monitoring oxygenation indices could be integrated into future multimodal assessments. For example, the arterial/alveolar oxygen tension (a/APO_2_) index (usually PaO_2_/[(FiO_2_ × 713) − 1.25 × PaCO_2_]) has been found to be a reliable and consistent index of respiratory impairment which depends on the amount of ventilation/perfusion mismatch and of shunting. Another oxygenation index, identified as SpO_2_/FiO_2_ ratio (SFR), highlights the discrepancies in supplemental oxygen requirements. For mechanically ventilated infants or those on noninvasive ventilation whose mean airway pressure (MAP) is determined, the oxygenation index O.I. (MAP × FiO_2_/PaO_2_) is of particular value for the early identification and grading of BPD-associated pulmonary hypertension.

## 8. Lung Oxygenation by Near-Infrared Spectroscopy in Preterm Infants

Near-infrared spectroscopy (NIRS) is examined below as a possible tool for the noninvasive, continuous lung monitoring of extremely preterm VLBW infants.

Regional saturation monitors or NIRS monitors are currently widely available in neonatal intensive care units. NIRS technology is a noninvasive tool very suitable for infants to achieve real-time tissue oxygenation painlessly with continuous monitoring (Appendix A). NIRS has been applied in various research and clinical care settings, but it has been mostly used to investigate solid organs, and the brain remains the most extensively studied one. In current neonatal care, continuous NIRS monitoring is applied to assess cerebral oxygenation and, to a lesser extent, renal, hepatic, and splanchnic oxygenation. Reference values for regional cerebral oxygen saturation have been published, and continuous NIRS monitoring of cerebral blood flow in preterm infants is an integral part of multiparametric brain monitoring. Nevertheless, it is a multi-organ monitoring tool that can also be applied to liquid samples and partially air-filled organs such as the lungs. Therefore, NIRS is being studied to assess its feasibility for new organs and tissues and to evaluate its role in clinical decision-making [41,42,43,44,45,46,47]. Studies on pulmonary NIRS, or lung NIRS, are very recent.

NIRS light penetration depth (1.0–1.5 cm) is adequate to assess lung parenchyma in preterm infants [48]. However, this application has been poorly investigated and data on normal values are lacking. Pulmonary oximetry via NIRS is expected to depend on pulmonary blood flow, as the bronchial circulation is estimated to receive only 1% of the left ventricular output. In a single-center study conducted in China, a population of 26 preterm infants (<32 weeks gestational age) underwent pulmonary NIRS monitoring, and lung regional oximetry (rSO_2_L) was positively correlated with arterial blood gas analysis values, both in the partial pressure of arterial oxygen (PaO_2_) and the arterial oxygen saturation (SaO_2_) [49]. Peripheral pulse oxygen saturation (SpO_2_) was also positively correlated with PaO_2_ and SaO_2_, but, interestingly, no significant correlation was found between rSO_2_ and SpO_2_ among preterm infants receiving oxygen therapy [49]. Hyperoxemia was defined as PaO_2_ > 100 mmHg while hypoxemia was defined as PaO_2_ < 80 mmHg. Lung oximetry (rSO_2_L) could be used better than SpO_2_ to predict both hyperoxemia and hypoxemia [49]. In conclusion, NIRS objectively reflected changes in oxygenation in lung tissue [49].

In preclinical research, oxygenation in the target tissue measured by NIRS has been shown to detect the effects of hypoxia earlier than pulse oximetry in the Yucatan miniature pig. A NIRS sensor may be used as an earlier detector of oxygen saturation changes in the routine clinical setting compared with the standard pulse oximeter [50].

In a prospective observational study conducted in Istanbul, Turkey, 110 healthy infants at >35 weeks’ gestation (87 at term and 23 late preterm infants) were studied for regional pulmonary oxygen saturations (rSO_2_L) and SpO_2_ immediately after birth [51]. Similarly to the SpO_2_ values, the rSO_2_L values increased between 3 and 5 min, and the postnatal increase reached a steady state around 9 min. Lung regional oximetry reflected the changes that occur during the transition from the foetal period to the neonatal period [51]. Lung NIRS values were significantly higher for babies born via caesarean section than for those born by vaginal delivery.

In the light of these findings, recently a single-centre prospective, observational, double-blind study of 40 late preterm and term infants diagnosed with neonatal respiratory disease was conducted in Turkey [52]. Lung rSO_2_ values by NIRS monitoring were measured during the 1st, 24th, 48th, and 72nd hours of hospitalisation in the NICU. The rSO_2_ values were significantly higher in infants with transient tachypnea (TTN) than in neonatal pneumonia. The authors proposed a cut-off value to distinguish the two neonatal diseases [52]. Furthermore, rSO_2_L values at the end of the first hour of life in the TTN group were very similar to the levels of a healthy newborn in the 3rd minute of life in the previous study. These data suggest that pulmonary NIRS levels reflect a prolonged transition period in infants with TTN. In addition, higher rSO_2_L values were inversely correlated with the required level of respiratory support. In a single-center prospective observational proof-of-concept study conducted in Florence, Italy, a cohort of 20 preterm infants with moderate respiratory distress syndrome (RDS) underwent continuous pulmonary NIRS monitoring, and rSO_2_L was found to have significant correlations with other oxygenation indices and with RDS severity [53].

It can be hypothesized that continuous pulmonary NIRS monitoring of lung oxygenation in preterm infants with evolving and established BPD is significantly different from reference values and, possibly, it is predictive of the diagnosis and severity of BPD. This has yet to be established, but there is a good basis for biological plausibility. Good quality studies are needed to support this new noninvasive continuous monitoring strategy. Local protocols and standard operating procedures are also needed for common use and better interpretation of available data and trends in new continuous monitoring techniques.

## 9. Conclusions

Noninvasive monitoring strategies for extremely preterm VLBW infants should be integrated into easy-to-use risk-scoring tools for objective, precise, and reliable BPD diagnosis at a very early stage, and as early predictors of treatment response.

Focusing on noninvasive monitoring systems for early postnatal assessment as early as the first days of life (0–72 h of life), we propose that the multimodal assessment of extremely preterm VLBW infants should include: 1. BPD risk stratification at birth, based on clinical and demographic data (the BPD risk-scoring tool published by Yu and colleagues or similar prediction models) [19]; 2. early analysis of biofluid biomarkers and “omic” signatures on umbilical cord blood, urine, and tracheal aspirate or bronchoalveolar lavage fluid samples collected at birth, and possibly repeated at 48–72 h of life [15]; 3. early interpretation of exhaled breath condensates, such as end-tidal carbon monoxide (ETCO) and fractional exhaled nitric oxide (FeNO) [15]; 4. chest radiography and lung ultrasound performed at admission and at 48–72 h of life, so as to interpret patterns and chest X-ray scores, such as the thoracic area (CRTA), and LUS scores [23,24,25,26,27,28,29,30]; 5. early detectable lung function parameters, such as standardized interpretation of flow volume loops in mechanical ventilation, or simplified pulmonary function tests (tidal flow-volume loops, adapted multiple breath washout, oscillometry, or others) to measure the functional residual capacity (FRC), the compliance (Crs), and the resistance of the respiratory system (Rrs) [36]; 6. individual performance in physiological tests that investigate the respiratory system response under stressful conditions, for example during weaning or decrease in respiratory support; 7. routine neonatal care data from respiratory monitoring, including peripheral pulse oximetry, blood gas analysis, transcutaneous PCO_2_ and PO_2_ (tcPCO_2_, tcPO_2_), and oxygenation indices such as SpO_2_/FiO_2_ ratio (SFR), arterial/alveolar oxygen tension (a/APO_2_), oxygenation index (O.I.), and arterial-alveolar O_2_ gradient (DAaO_2_); 8. new noninvasive, continuous pulmonary monitoring strategies. We proposed continuous pulmonary NIRS monitoring as an innovative technique to investigate the small lungs of premature infants (Figure 2).

We believe that none of these tools alone can provide effective answers, but that the integration of these data will effectively support clinical decision-making. Early indicators of treatment response could be deduced from repetition of the same investigations at standardized time points (e.g., admission, 48–72 h of life, 7 days of life, 14 days of life, 28 days of life, 36 weeks of gestational age, 40 weeks of gestational age, and post-discharge).

## Figures and Tables

**Figure 1 children-10-01753-f001:**
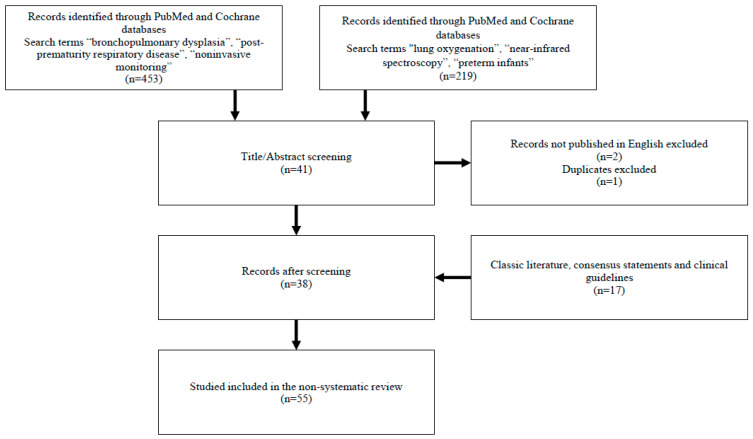
Sources and selection criteria.

**Figure 2 children-10-01753-f002:**
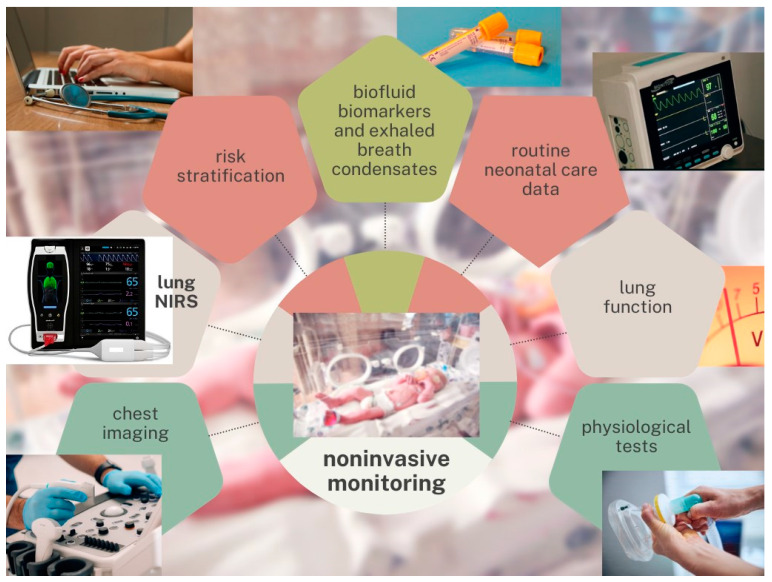
Noninvasive monitoring strategies. Abbreviation. NIRS: near-infrared spectroscopy. Note: We propose noninvasive monitoring strategies to predict bronchopulmonary dysplasia (BPD) at a very early stage in extremely preterm very-low-birth-weight infants. Multimodal assessments should be integrated into easy-to-use risk-scoring tools to support clinical decision-making.

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
