# Peer review of "Noninvasive Monitoring Strategies for Bronchopulmonary Dysplasia or Post-Prematurity Respiratory Disease: Current Challenges and Future Prospects"

_children, 2023, doi:10.3390/children10111753_

Round 1
Reviewer 1 Report
Comments and Suggestions for Authors
In this non-systematic review, the authors presented an overview of strategies for monitoring preterm infants with early and evolving bronchopulmonary dysplasia (BPD)-post-prematurity respiratory disease (PPRD), and made some remarks on future prospects, with a focus on near-infrared spectroscopy (NIRS). Overall, this review provides incremental insights of noninvasive monitoring strategies for BPD or PPRD. But there are some concerns should be addressed before it is considered for publication.
1. The authors need to pay attention to the innovation of the cited literatures in the review, and the literatures were mainly published in the last five years, unless they were classic literatures.
2. To improve the quality of this review, it is suggested to add several figures or tables to cover important information.
3. Please add a scetion “The literature searched strategy”.
4. In section 4, there are four subheadings, but their serial numbers are all wrong.
Reviewer 2 Report
Comments and Suggestions for Authors
Dear editors,
Thank you very much for the opportunity to review this interesting article.
The authors deal with an interesting topic and one that is never old. Despite the therapeutic advances in recent years, pulmonary broncho-dysplasia remains a significant health problem that generates a high personal, social, and economic cost in the environment of those who suffer from it.
Although the main drawback that may arise concerning this work is the non-systematic nature of the review, this design decision corresponds to the authors. The result is an interesting work that, although it does not respond to a systematic approach, does update an essential field in which it is relevant to note updates such as the one presented in this review.
Therefore, the work submitted may be of considerable interest, although minor revisions may improve the article's readability. These are the following:
Although the authors do not follow a reproducible methodology, it would help readers if they briefly mentioned how they developed their review.
It would also be helpful to contextualize the strategies presented more, perhaps by mentioning or justifying them better in the introduction.
The same thing happens in the first paragraph and section 2.1: in both places, one reference is cited several times (reference 1 in the first paragraph and reference 19 in section 2.1). This is strange to read. It would be helpful if the authors clarified that the aspects cited in these texts correspond to a single reference so they do not have to keep repeating the same reference in different citations.
In section 3, when the authors mention biomarkers and "omic" signatures, they could devote one or two lines to developing these concepts.
It would be helpful if the authors presented some conclusions in this same section as in other sections.
Given that the central theme of the article seems interesting and relevant to the clinic, I believe that the work developed by the authors would allow the development of more elaborate, more extensive conclusions so as not to remain in the simple recommendation that these techniques should be included in the monitoring strategies. Although this conclusion should be reflected (probably at the end of the text), it is implicitly included throughout the previous text. For this reason, when the reader arrives at this concluding paragraph, they expect a more integrated summary of the different conclusions reached by the authors in each of the sections presented in addition to the conclusion.
Round 2
Reviewer 1 Report
Comments and Suggestions for Authors
The author lacks strict scientific research training, and can't even revise and reply to reviewers' comments in a standardized manner
Author Response
Thank you for taking the time to review this manuscript.